# The partition representation of enzymatic reaction networks and its application for searching bi-stable reaction systems

**Takashi Naka**⦿*

Faculty of Science and Engineering, Kyushu Sangyo University, Fukuoka, Japan

* naka@is.kyusan-u.ac.jp

**Data Availability Statement:** All relevant data are within the manuscript and its Supporting Information files.

**Funding:** The author(s) received no specific funding for this work.

## Abstract

The signal transduction system, which is known as a regulatory mechanism for biochemical reaction systems in the cell, has been the subject of intensive research in recent years, and its design methods have become necessary from the viewpoint of synthetic biology. We proposed the partition representation of enzymatic reaction networks consisting of post-translational modification reactions such as phosphorylation, which is an important basic component of signal transduction systems, and attempted to find enzymatic reaction networks with bistability to demonstrate the effectiveness of the proposed representation method. The partition modifiers can be naturally introduced into the partition representation of enzymatic reaction networks when applied to search. By randomly applying the partition modifiers as appropriate, we searched for bistable and resettable enzymatic reaction networks consisting of four post-translational modification reactions. The proposed search algorithm worked well and we were able to find various bistable enzymatic reaction networks, including a typical bistable enzymatic reaction network with positive auto-feedbacks and mutually negative regulations. Since the search algorithm is divided into an evaluation function specific to the characteristics of the enzymatic reaction network to be searched and an independent algorithm part, it may be applied to search for dynamic properties such as biochemical adaptation, the ability to reset the biochemical state after responding to a stimulus, by replacing the evaluation function with one for other characteristics.

## Introduction

The intracellular signal transduction system functions as a mechanism to control cell proliferation, apoptosis, differentiation, and homeostasis, and its malfunction is thought to be one of the substantial causes of cancer formation for cells. The regulatory mechanism of signal transduction systems is closely related to the mechanism of action of drugs, such as anticancer drugs, and has therefore been the subject of extensive research in recent years. In addition, from the viewpoint of synthetic biology, which has been the focus of much research in recent years, design methods for signal transduction systems have become necessary. However, because biochemical reactions are nonlinear, it is difficult to establish theoretical analysis

**Competing interests:** The authors have declared that no competing interests exist.

methods, and signal transduction systems in particular have not been systematically analyzed because their interactions are more complex than those of intracellular metabolic systems, which they control. The current situation is that the parameters are fixed for each system to be analyzed, and the analysis is performed by computer simulation.

The MAPK cascade, a representative signal transduction system, which relays cell growth factor (EGF) from the cell membrane to the cell nucleus, and its abnormality is thought to be the cause of cell canceration, and much knowledge has been obtained [1–6]. A major component of the signal transduction system represented by the MAPK cascade is the cyclic reaction system, which activates and deactivates enzymes by phosphorylation and dephosphorylation. The cyclic reaction system is a combination of two post-translational modification reactions. Therefore, we have been conducting a comprehensive analysis of the stability of regulatory networks that consist of activation and inactivation cyclic reaction systems of enzymes and their mutual controls [7]. There, the control relationship between the cyclic reaction systems is represented by a control matrix. The control matrix, which is an adjacency matrix, has 1 as its $i$-th row and $j$-th column components when the activating enzyme of cyclic reaction system $j$ catalyzes the activation of cycle reaction system $i$, and -1 when it catalyzes the deactivation.

In a similar study, Kuwahara et al. [8] exhaustively analyzed the influence of the control structure on the stochastic properties for regulatory networks of three to five nodes with only one feedback control. They also mentioned the control structures in which bistability emerges. Ramakrishnan et al. [9], Shah et al. [10], and Siegal-Gaskins et al. [11] have conducted exhaustive analyses of the ultrasensitive properties and bistability of signaling systems. Ultrasensitivity was reported as a response characteristic of the MAPK cascade, a typical signaling system mentioned above [1]. In a system with ultrasensitivity, the steady-state enzyme concentration of the output chemical species with respect to the enzyme concentration of the input chemical species increases rapidly after a certain threshold of the input. Bistability is when this ultrasensitivity is more strongly expressed, and hysteresis appears in the change of output to input. Thus, in a region with input, two steady state values of the output appear. In particular, it is said to be resettable if it has the property of being able to mutually transition between two steady states by varying the total concentration of the input chemical species within the range of protein concentrations expected in the cell [12].

Ma et al. [13] and Yao et al. [14] analyzed the biochemical adaptation and robustness properties using a similar control network of three nodes. Biochemical adaptation is the property that when the concentration of an input chemical species is increased in a steady-state system, the response of the output chemical species transiently increases and then decreases to a value close to the original steady-state value [13]. Robustness is the property that the various response properties of a system are less sensitive to the concentration of other chemical species in the system [12].

Adler et al. [15] focused on fold-change detection, which is a dynamic property in which the temporal pattern of the system's output in response to a transient input change in the cell depends on the ratio rather than the difference of the input changes, demonstrating the effectiveness of comprehensive analysis using regulatory networks. In these studies, the enzymatic reaction mechanism is mainly based on the Michaelis-Menten approximation or a simplified linear equation in order to reduce the computational complexity. However, it has been reported that approximations can lead to inaccuracies in the bistability and dynamics of the system [16, 17]. In order to accurately analyze the properties of a system, it would be better to describe the system using only the law of mass action, but this would increase the number of parameters of the system and make the originally huge search space even larger, making exhaustive analysis difficult to carry out. Therefore, there is a need to develop a method to search for enzymatic reaction networks with useful properties such as ultrasensitivity,

bistability, biochemical adaptation, robustness, and fold-change detection as mentioned above [18–20].

In this study, we focus on the enzymatic reaction network, which is formulated as a regulatory network in which post-translational modification reactions are the elements and they mutually regulate each other. Post-translational modification reactions, such as phosphorylation, are smaller units than the cyclic reaction systems. Instead of the control matrix described above, we propose to use a partition of the set as its representation, which is considered to be more suitable for search. In addition, we report the results of our search for a bistable and resettable enzymatic reaction network with the aim of demonstrating its effectiveness. Resettable is the property of being able to transition between two steady states by changing the total concentration of the input chemical species within the range of expected protein concentrations in the cell [12].

## Materials and methods

### Partition representations for enzymatic reaction networks

A post-translational modification reaction with Michaelis-Menten-type enzymatic reaction as the reaction mechanism is considered as the basic component, considering to construct the enzymatic reaction network composed of $N$ components. In this regard, reactions in which multiple enzymes bind to a single substrate simultaneously are not considered for simplicity in this formulation. The Michaelis-Menten-type enzymatic reaction is a series of chemical reactions in which an enzyme (E) binds to a substrate (S) to form a temporary enzyme-substrate complex (C), the substrate is transformed into a product (P) on the complex, and then decomposes into the product (P) and the enzyme (E), as shown in Eq (1), where the reaction rate constants are denoted by $a$, $d$, and $k$.

$$S + E \underset{d}{\overset{a}{\rightleftarrows}} C \overset{k}{\rightarrow} P + E \tag{1}$$

The substrates, enzymes, and products of $N$ Michaelis-Menten-type enzymatic reactions are denoted by $S_i$, $E_i$, and $P_i$, respectively, and the set is denoted by $M$. That is, $M = \{S_1, E_1, P_1, S_2, E_2, P_2, \ldots, S_N, E_N, P_N\}$.

Take one partition $P$ of the set $M$. A partition is a family of mutually disjoint subsets of a set, and the union of those subsets is the original set. By identifying the elements of the partition, i.e., the chemical species that belong to one subset of the original set $M$, an enzymatic reaction network is constructed from multiple post-translational modification reactions. Fig 1 shows an example of a basic enzymatic reaction network consisting of post-translational modification reactions. The primary building blocks of post-translational modification reactions are numbered consecutively; the substrate, enzyme, and product of the $i$-th post-translational modification reaction are labeled $S_i$, $E_i$, and $P_i$, respectively. If the chemical species to be equated is the substrate $S_i$ and the product $P_i$, the names of the species to be equated are listed in the rectangle. The enzyme $E_i$ is represented by the small circle between the red arrows, and if there is a chemical species that is identical to the enzyme, it is connected to that chemical species by a dotted line. It may be easier to understand intuitively if you think of the small circle as representing the enzyme-substrate complex.

Fig 1A–1D are examples of an enzymatic reaction network consisting of $N$ = 2, i.e., two post-translational modification reactions, where the original set of partitions is $M = \{S_1, E_1, P_1, S_2, E_2, P_2\}$. Fig 1A shows the enzymatic reaction network represented by the partition $\{\{S_1\}, \{E_1\}, \{P_1, E_2\}, \{S_2\}, \{P_2\}\}$ of $M$, where the product of enzymatic reaction 1 functions as the enzyme in enzymatic reaction 2. The identification of $P_i$ and $E_j$ corresponds to the fact that the

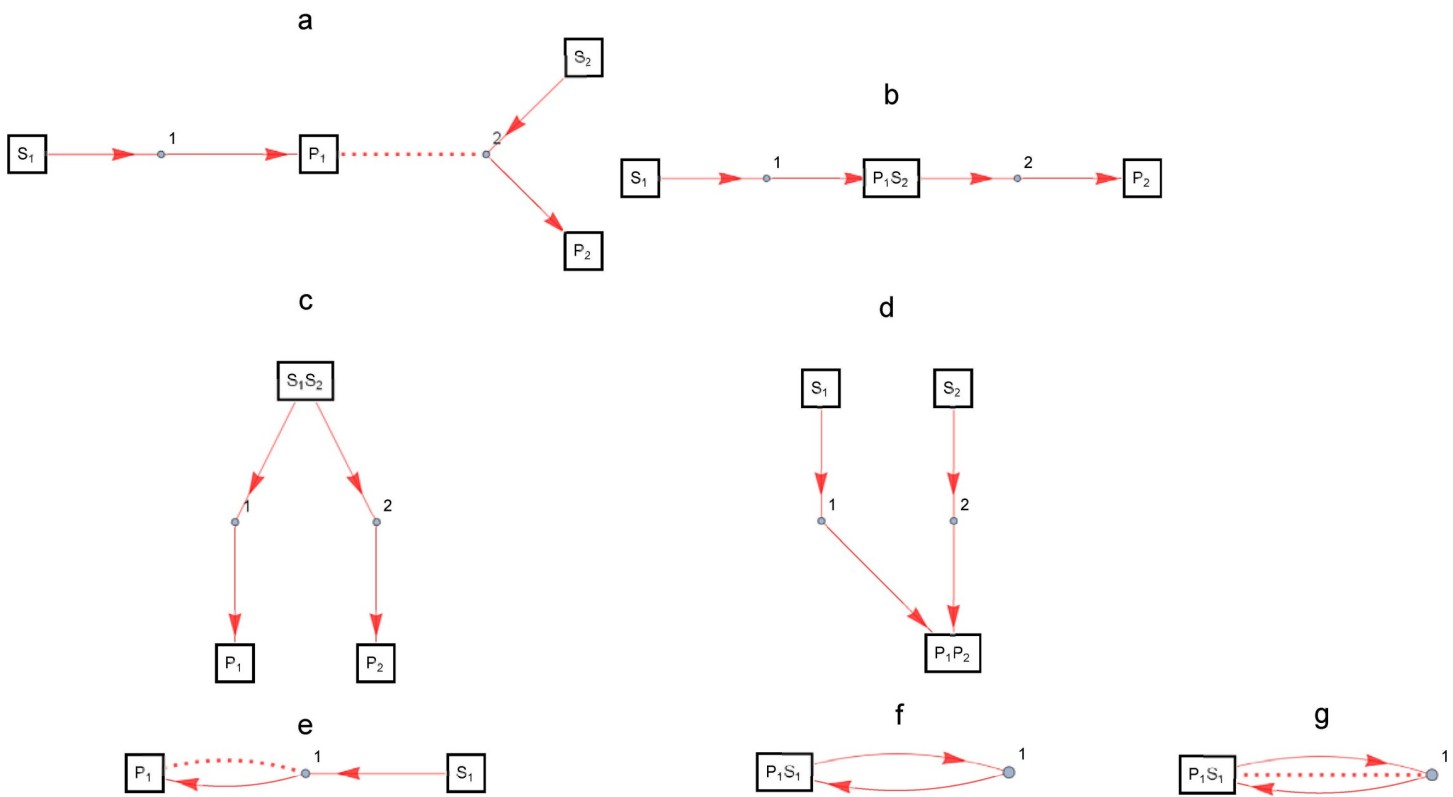

**Fig 1. Basic enzymatic reaction networks consisting of post-translational modification reactions.** Examples of enzymatic reaction networks composed of post-translational modification reactions as the primary building blocks; a: product of enzymatic reaction 1 catalyzes enzymatic reaction 2, b: two-step reaction, c: branching reaction, d: confluence reaction, e: autocatalysis, f: association-dissociation reaction, g: dimer formation separation reaction. The primary building blocks of post-translational modification reactions are numbered consecutively; the substrate, enzyme, and product of the $i$-th post-translational modification reaction are labeled $S_i$, $E_i$, and $P_i$, respectively. If the chemical species to be equated is the substrate $S_i$ and the product $P_i$, the names of the species to be equated are listed in the rectangle. The enzyme $E_i$ is represented by the small circle between the red arrows, and if there is a chemical species that identifies with the enzyme, it is connected to that species by a dotted line.

product of enzymatic reaction $i$ functions as the enzyme of enzymatic reaction $j$. This is a typical mode of enzymatic reaction chain in which the activated enzyme catalyzes other enzymatic reactions as seen in signal transduction systems. Fig 1B shows the enzymatic reaction network represented by the partition $\{\{S_1\}, \{E_1\}, \{P_1, S_2\}, \{E_2\}, \{P_2\}\}$ of $M$, where the products of enzymatic reaction 1 are the substrates of enzymatic reaction 2. The identification of $P_i$ and $S_j$ is a representation of the two-step enzymatic reaction from $S_i$ to $P_j$. Fig 1C and 1D show the enzymatic reaction networks represented by the partitions, $\{\{S_1, S_2\}, \{E_1\}, \{P_1\}, \{E_2\}, \{P_2\}\}$ and $\{\{S_1\}, \{E_1\}, \{S_2\}, \{E_2\}, \{P_1, P_2\}\}$ of $M$, respectively. The identification of $S_i$ and $S_j$, and $P_i$ and $P_j$, respectively, is a representation of the branching into and confluence from two enzymatic reactions.

Fig 1E to g are examples of enzymatic reaction networks consisting of $N = 1$, i.e., one post-translational modification reaction, where the original set of partitions is $M = \{S_1, E_1, P_1\}$. Fig 1E shows the enzymatic reaction network represented by this partition $\{\{S_1\}, \{E_1, P_1\}\}$ of $M$, which is a representation of an autocatalytic reaction. It corresponds to the case where $i = j = 1$ in Fig 1A. Fig 1F shows the enzymatic reaction network represented by the partition $\{\{S_1, P_1\}, \{E_1\}\}$ of $M$, showing the association and dissociation reaction between the identical substrate and product $S_1 = P_1$ and the enzyme $E_1$. The identification of $S_i$ and $P_i$ is a representation of the association and dissociation reaction with the enzyme. Fig 1G shows the enzymatic reaction network represented by the partition $\{\{S_1, P_1, E_1\}\}$ of $M$, where the identified substrate

and product and the enzyme $S_1 = P_1 = E_1$ associate and dissociate themselves, corresponding to the dimer formation separation reaction. The identification of $S_i$, $P_i$, and $E_i$ is an expression of the dimer formation-separation reaction.

Fig 2 shows examples of more complex enzymatic reaction networks and their partition representations. Fig 2A shows an example of an enzymatic reaction network consisting of $N = 4$, i.e., four post-translational modification reactions, where the original set of the partition is $M = \{S_1, E_1, P_1, S_2, E_2, P_2, S_3, E_3, P_3, S_4, E_4, P_4\}$ and the partitions is $\{\{S_1, P_2\}, \{P_1, S_2, E_1, E_4\}, \{S_3, P_4\}, \{P_3, S_4, E_2, E_3\}\}$. This is an example of an enzymatic reaction network consisting of two cyclic reaction systems with auto-activating and mutually inhibitory feedbacks, which is an typical of enzymatic reaction network with bistable properties. The MAPK cascade, one of the representative signaling systems, is shown in Fig 2B. The MAPK cascade consists of a cascade of three kinases, MAPKKK, MAPKKK, and MAPK. The activated enzyme activates the phosphotransferase in the next step, thereby transmitting signals within the cell. An example of an enzymatic reaction network consisting of 10 post-translational modification reactions, with $M = \{S_1, E_1, P_1, S_2, E_2, P_2, \ldots, S_{10}, E_{10}, P_{10}\}$ can be represented by the partition $\{\{P_2, S_1\}, \{P_1, S_2, E_3, E_4\}, \{E_1\}, \{E_2\}, \{P_6, S_3\}, \{P_3, P_5, S_4, S_6\}, \{P_4, S_5, E_7, E_8\}, \{P_{10}, S_7\}, \{P_7, P_9, S_8, S_{10}\}, \{P_8, S_9\}, \{E_5\}, \{E_6\}, \{E_9\}, \{E_{10}\}\}$, where $\{P_2, S_1\}$ and $\{P_1, S_2, E_3, E_4\}$ are the inactive and active MAPKKKs, respectively. Also, $\{P_6, S_3\}$, $\{P_3, P_5, S_4, S_6\}$, and $\{P_4, S_5, E_7, E_8\}$ are inactive, one phosphorylated, and two phosphorylated MAPKKs, respectively. $\{P_{10}, S_7\}$, $\{P_7, P_9, S_8, S_{10}\}$, and $\{P_8, S_9\}$ are inactive, one-phosphorylated, and two-phosphorylated MAPKs, respectively.

The advantage of the partition representation is that every partition corresponds to an enzymatic reaction network, and thus there is a one-to-one relationship between the partition and

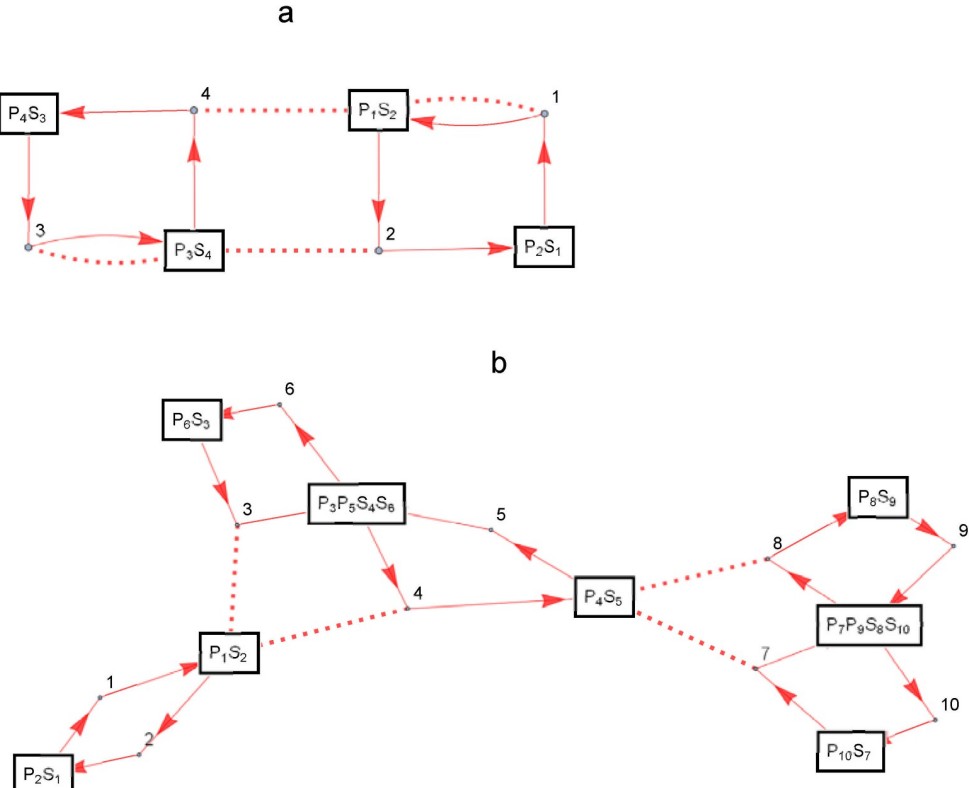

**Fig 2. Complex enzymatic reaction networks consisting of post-translational modification reactions.** The way to represent the figure is the same as in Fig 1; a: auto-activating mutual inhibitiory network, b: MAPK cascade.

the enzymatic reaction network. The number of different enzymatic reaction networks consisting of $N$ post-translational modification reactions is the same as the total number of different partitions of the set with number of elements $3N$, a number whose value is known as the Bell number $B_{3N}$, expressed in the recurrence equation as in Eq (2), where $B_0 = B_1 = 1$.

$$B_{3N+1} = \sum_{k=0}^{3N} {}_{3N}C_k B_{3N} \tag{2}$$

For example, the total number of enzymatic reaction networks consisting of four post-translational modification reactions, as shown in Fig 2A, is $B_{12} = 4213597$.

In the space where the partition is an element, we can introduce the distance $D(A, B)$ between the partition $A = \{a_1, a_2, \ldots, a_m\}$ and the partition $B = \{b_1, b_2, \ldots, b_n\}$ of the set $M$ as shown in Eq (3). Here, $a_i$ and $b_j$ are subsets of the set $M$.

$$s(a_i, b_j) = \#(a_i \cup b_j - a_i \cap b_j), \; d(A, b_j) = \min_i s(a_i, b_j)$$
$$D(A, B) = \max(\max_j d(A, b_j), \; \max_i d(B, a_j)) \tag{3}$$

The symbol  is a function that returns the number of elements in a set. Therefore, $\#(a_i \cup b_j - a_i \cap b_j)$ is the number of elements that are not common to the sets $a_i$ and $b_j$. For example, let the source set of the partition be $M = \{x_1, x_2, x_3, x_4, x_5, x_6\}$, and consider three partitions, $A = \{a_1, a_2, a_3\}$, $B = \{b_1, b_2\}$, and $C = \{c_1, c_2\}$, where $a_1 = \{x_1, x_2, x_3\}$, $a_2 = \{x_4, x_5\}$, $a_3 = \{x_6\}$, $b_1 = \{x_1, x_2\}$, $b_2 = \{x_3, x_4, x_5, x_6\}$, $c_1 = \{x_1, x_2, x_3\}$, $c_2 = \{x_4, x_5, x_6\}$. Then $d(A, b_1) = \min_i s(a_i, b_1) = s(a_1, b_1) = 1$, $d(A, b_2) = \min_i s(a_i, b_2) = s(a_2, b_2) = 2$, so $\max_j d(A, b_j) = d(A, b_2) = 2$, and similarly $\max_j d(B, a_j) = 3$. Therefore, $D(A, B) = \max(2, 3) = 3$. Similarly, we can see that $D(B, C) = 1$ and $D(A, C) = 2$. In other words, $A$ and $B$ are the farthest, $B$ and $C$ are the closest, and $A$ and $C$ are in the middle distance. To check this in an enzymatic reaction network, for example, let the original set of chemical species be $M = \{S_1, P_2, E_2, P_1, E_1, S_2\}$. In other words, $x_1 = S_1$, $x_2 = P_2$, $x_3 = E_2$, $x_4 = P_1$, $x_5 = E_1$, $x_6 = S_2$. Then partition $A$ becomes $\{\{S_1, P_2, E_2\}, \{P_1, E_1\}, \{S_2\}\}$, $B$ becomes $\{\{S_1, P_2\}, \{E_2, P_1, E_1, S_2\}\}$, and $C$ becomes $\{\{S_1, P_2, E_2\}, \{P_1, E_1, S_2\}\}$, corresponding to a, b, and c of the enzymatic reaction network shown in Fig 3.

It can be seen that the similarity of the enzyme reaction network correlates with this distance. Furthermore, this distance between partitions satisfies the triangle inequality and can be used to visualize the distribution of partitions.

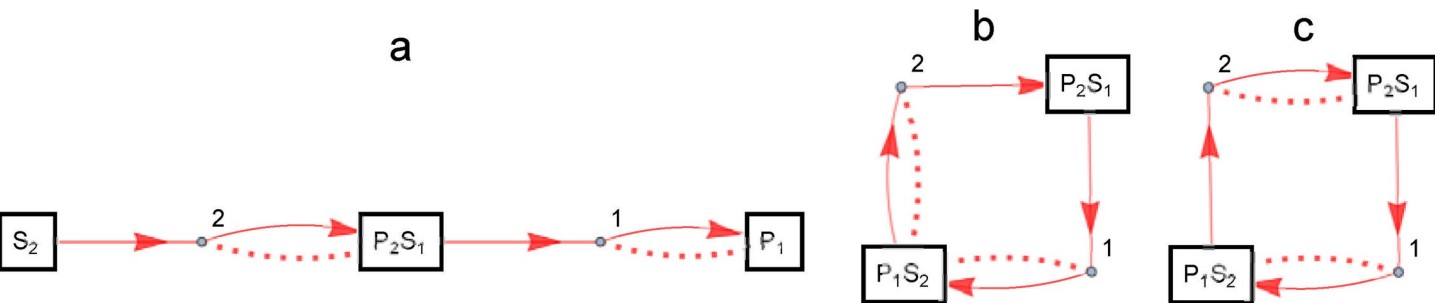

**Fig 3. Enzymatic reaction networks specified by divisions and distances between the divisions.** The way to represent the figure is the same as in Fig 1. The enzymatic reaction networks represented by the partitions; A: $\{\{S_1, P_2, E_2\}, \{P_1, E_1\}, \{S_2\}\}$, B: $\{\{S_1, P_2\}, \{E_2, P_1, E_1, S2\}\}$, C: $\{\{S_1, P_2, E_2\}, \{P_1, E_1, S_2\}\}$.

### Derivation of differential equation systems and conservation laws from partitions

From the partition, we can derive a system of differential equations and conservation laws that describe the behavior of the corresponding enzymatic reaction network as follows. If the number of post-translational modification reactions constituting the enzymatic reaction network is $N$, the total number of elements in the set from which the partitioning is made is $3N$, since three elements, that is, a substrate, an enzyme, and a product, correspond to one post-translational modification reaction. Therefore, if we denote the elements by $x_i$, the source set of the partition can be expressed as $M = \{x_1, x_2, \ldots, x_{3N}\}$.

Let one of the partitions of $M$ be $A = \{a_1, a_2, \ldots, a_m\}$, and let $s_k$ be the set of subscript $i$ of $x_i$ which is an element of the $k$th element $a_k$ of the partitions $A$. For this partition $A$, a system of differential equations can be generated by the following procedure.

1. Generate $4N$ reaction rate equations for substrate $S_i$, enzyme $E_i$, enzyme-substrate complex $C_i$, and product $P_i$ in each post-translational modification reaction, as shown in Eq (4). Note that $x_{3N+1}, x_{3N+2}, \ldots, x_{4N}$ correspond to the enzyme-substrate complex $C_i$.

$$\dot{x}_j = f_j(x_1, \cdots, x_{4N}), \quad j = 1, \cdots, 4N \tag{4}$$

2. For each element $a_k$ in the partition, generate a chemical species $y_k$ that identifies the all elements of $a_k$, and replace the original reaction rate equations for the elements of $a_k$ with the following Eq (5), while all elements of $a_k$ appearing on the right side of Eq (5) are replaced by $y_k$. The $s_k$ is the subscript set of the elements of $a_k$.

$$\dot{y}_k = \sum_{j \in s_k} f_j(x_1, \cdots, x_{4N}), \quad k = 1, \cdots, m \tag{5}$$

For the derivation of Eq (4), if only the law of mass action is used as the reaction equation for the post-translational modification reaction, and the variable name is expressed as $x$[chemical species], the reaction equation derived is the following for the post-translational modification reaction $i$.

$$
\begin{aligned}
\dot{x}[S_i] &= -a_i x[S_i] x[E_i] + d_i x[C_i] \\
\dot{x}[E_i] &= -a_i x[S_i] x[E_i] + (d_i + k_i) x[C_i] \\
\dot{x}[C_i] &= a_i x[S_i] x[E_i] - (d_i + k_i) x[C_i] \\
\dot{x}[P_i] &= k_i x[C_i]
\end{aligned}
\tag{6}
$$

For example, if the original set of partitions is $M = \{S_1, E_1, P_1, S_2, E_2, P_2\}$, and the partition is $A = \{\{S_1\}, \{E_1\}, \{P_1, E_2\}, \{S_2\}, \{P_2\}\}$ as seen in Fig 1A, then by applying the law of mass action, the above Eq (4) becomes the following Eq (7), where the variable name is represented by $x$

[chemical species].

$$
\begin{aligned}
\dot{x}[S_1] &= -a_1 x[S_1] x[E_1] + d_1 x[C_1] \\
\dot{x}[E_1] &= -a_1 x[S_1] x[E_1] + (d_1 + k_1) x[C_1] \\
\dot{x}[C_1] &= a_1 x[S_1] x[E_1] - (d_1 + k_1) x[C_1] \\
\dot{x}[P_1] &= k_1 x[C_1] \\
\dot{x}[S_2] &= -a_2 x[S_2] x[E_2] + d_2 x[C_2] \\
\dot{x}[E_2] &= -a_2 x[S_2] x[E_2] + (d_2 + k_2) x[C_2] \\
\dot{x}[C_2] &= a_2 x[S_2] x[E_2] - (d_2 + k_2) x[C_2] \\
\dot{x}[P_2] &= k_2 x[C_2]
\end{aligned}
\tag{7}
$$

The equations corresponding to Eq (5), obtained by equating $P_1$ and $E_2$ according to division $A$, are Eq (8). In particular, when some chemical species are considered identical, they are listed in brackets.

$$
\begin{aligned}
\dot{x}[S_1] &= -a_1 x[S_1] x[E_1] + d_1 x[C_1] \\
\dot{x}[E_1] &= -a_1 x[S_1] x[E_1] + (d_1 + k_1) x[C_1] \\
\dot{x}[C_1] &= a_1 x[S_1] x[E_1] - (d_1 + k_1) x[C_1] \\
\dot{x}[P_1, E_2] &= k_1 x[C_1] - a_2 x[S_2] x[P_1, E_2] + (d_2 + k_2) x[C_2] \\
\dot{x}[S_2] &= -a_2 x[S_2] x[P_1, E_2] + d_2 x[C_2] \\
\dot{x}[C_2] &= a_2 x[S_2] x[P_1, E_2] - (d_2 + k_2) x[C_2] \\
\dot{x}[P_2] &= k_2 x[C_2]
\end{aligned}
\tag{8}
$$

For the conservation law, we first generate a set of lists $Q = \{q_1, \ldots, q_i, \ldots, q_{2N}\}$ of chemical species whose total concentration is conserved for each post-translational modification reaction. The reason why the total number is $2N$ is that for each post-translational modification reaction, there is a conservation law for the enzyme and a conservation law for the substrate. Next, the following procedure is applied sequentially to each element $a_k$ of the partition $A$, in turn.

1. Concatenate all the list of elements in $Q$ that contain elements in common with $a_k$, and let $r_k$ be the list.

2. Remove the elements of $a_k$ from the list rk and add the chemical species $y_k$, which is identical to the elements of $a_k$.

3. Add $r_k$ to the element of $Q$ that does not contain an element in common with $a_k$, and make it $Q$ again.

It is important to note that the above procedure concatenates, rather than sums, the elements of $Q$ that contain elements in common with $a_k$. This is because the overlap is intrinsic when the list to be combined contains the common enzyme-substrate complex $C_i$.

If only the law of mass action is used as the reaction equation for the post-translational modification reaction, and the variable name is represented by $x$[chemical species], the initial elements of the set of conservation laws $Q$ to be derived are the following for the post-translational modification reaction $i$.

$$
\{x[S_i], x[C_i], x[P_i]\}, \{x[E_i], x[C_i]\}
\tag{9}
$$

For example, for the enzymatic reaction networks in Fig 1A, the partition is $A = \{\{S_1\}, \{E_1\},$

$\{P_1, E_2\}, \{S_2\}, \{P_2\}\}$. Then, if the variable names are denoted by $x$[chemical species] as in the example above, the original list of conservation laws becomes $\{\{x[S_1], x[C_1], x[P_1]\}, \{x[E_1], x[C_1]\}, \{x[S_2], x[C_2], x[P_2]\}, \{x[E_2], x[C_2]\}\}$, and by equating $P_1$ and $E_2$, we get $Q = \{\{x[S_1], x[C_1], x[P_1, E_2], x[C_2]\}, \{x[E_1], x[C_1]\}, \{x[S_2], x[C_2], x[P_2]\}\}$.

### Resettabe bistability

A search for bistable and resettable enzymatic reaction networks was attempted to demonstrate the effectiveness of the partition representation of enzymatic reaction networks. Resettable is the property of being able to mutually transition between two steady states by varying the total concentration of the input chemical species within the range of expected protein concentrations in the cell [12]. A typical resettable bistable enzymatic reaction network is shown in Fig 2A.

The relationship between bistability and resettability when the input chemical species is $E_1$ and the output chemical species is $P_2$ is shown in Fig 4. For example, in the enzymatic reaction network of Fig 2A, the input chemical species $E_1$ corresponds to the small circle labeled 1, and the output chemical species $P_2$ corresponds to the rectangle in the lower left corner that equates $S_1$ and $P_2$. The horizontal axis is the logarithm of the input species concentration with a base of 2, and the vertical axis is the total output species concentration normalized to between 0 and 1. The range of protein concentrations expected in the cell is $2^{-5}$ to $2^5$ in m mol/m$^3$.

The curve represents the concentration of the output species at equilibrium, where the solid line corresponds to the stable equilibrium point, or steady state value, and the dashed line corresponds to the unstable equilibrium point.

Fig 4A shows the resettable bistable aspect. The solid blue curves at both ends of the graph correspond to a single steady state, which are monostable states. The red curve in the center has two steady states, corresponding to bistable states. As the total concentration of the input species is increased from the smaller one, the black arrow on the right makes a discontinuous jump from the upper steady state to the blue monostable state on the right. When the total input species concentration is reduced from that state, the system jumps to the steady state below at the black arrow on the left side. Thus, it is a resettable bistable because it can mutually transition between two steady states.

Fig 4B is an example where the curve in Fig 4A extends to the right and the region of monostability on the right exceeds the range of protein concentrations expected in the cell. In this case, the black arrows show that it is possible to jump from the upper steady state to the lower steady state, but it is not possible to jump from the lower steady state to the upper steady state, even if the total input chemical species concentration is increased within the range of expected protein concentrations in the cell. This is an example of a bistable that is not resettable.

Fig 4C shows an example where the graph extends further to the left. In such a case, if the initial state of the system is above the red dotted line in the center, it shifts to the upper steady state, and if it is below, it shifts to the lower steady state, and the state cannot be changed even if the total concentration of the input chemical species is changed within the range of the expected protein concentration in the cell. This is another example of bistability that is not resettable.

### Exploration of enzymatic reaction networks in the partition representation space

For the partition representation of the enzymatic reaction network, an algorithm to search for the enzymatic reaction network with the desired response characteristics can be implemented in a natural way by sequentially applying the following two partition modifiers.

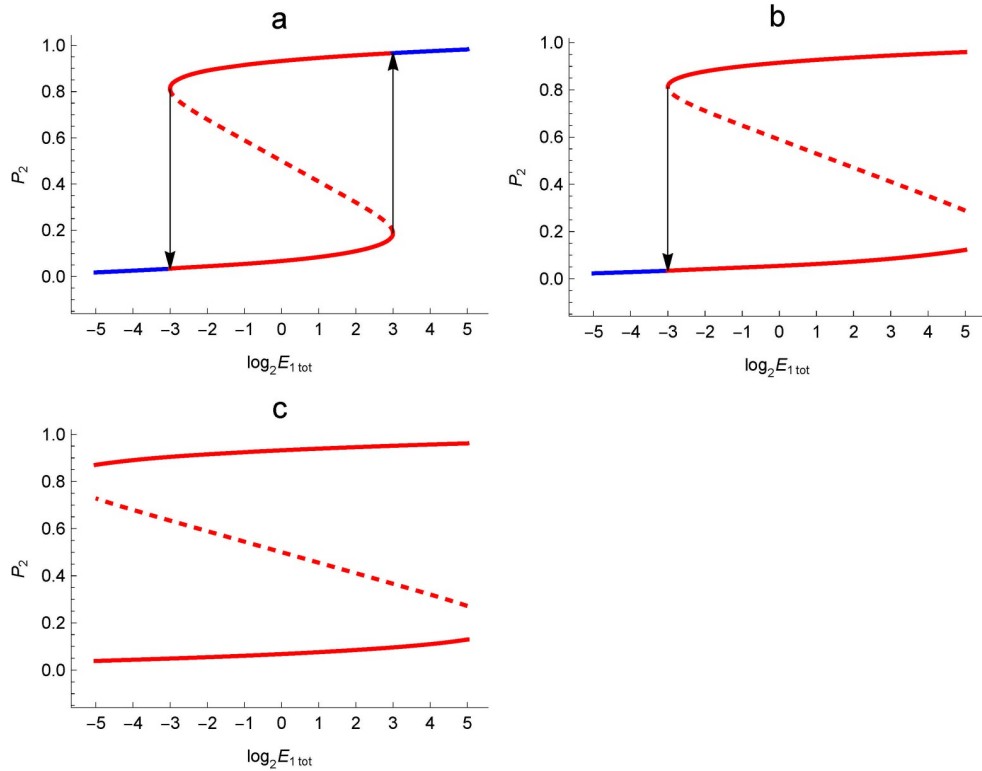

**Fig 4. Relationship between bistability and resettability.** The horizontal axis is the total concentration of the enzyme including the input chemical species $E_1$ and the vertical axis is the relative concentration of the output chemical species $P_2$ to the total concentration; a: resettable bistable, b: non-resettable bistable, but the transition from the upper stable state to the lower can occur, c: non-resettable bistable. The solid line corresponds to the stable equilibrium point, i.e., the steady state value, and the dashed line corresponds to the unstable equilibrium point, The solid blue lines at both ends depict the monostable state.

- Separate partition modifier: Randomly selects an element from a partition and randomly separates it into two elements. For example, by separating the second element $\{P_1, E_1, S_2\}$ of the partition $\{\{S_1, P_2, E_2\}, \{P_1, E_1, S_2\}\}$ corresponding to the enzymatic reaction network C shown in Fig 3 into two elements $\{P_1, E_1\}$ and $\{S_2\}$, we obtain the partition $\{\{S_1, P_2, E_2\}, \{P_1, E_1\}, \{S_2\}\}$ corresponding to the enzymatic reaction network A is obtained.

- Join partition modifier: Combines two randomly selected elements. For example, in the example of Fig 3 above, by combining the second and third elements of the partition corresponding to the enzymatic reaction network A, the partition corresponding to the enzymatic reaction network C is obtained.

First, we set up an evaluation function $\phi(P,v)$ that quantifies the desired response properties of the dynamics or steady state of the enzymatic reaction network generated from the partition. $\phi(P,v)$ is a function of the partition $P$, with the parameter $v$, which is a pair of reaction rate constants of each post-translational modification reaction and the total concentration of enzymes that constitute the enzymatic reaction network corresponding to the partition.

In the following algorithm, we use the function $\Phi(P)$ to perform a random search on the parameter value $v$ for a given partition $P$. The function $\Phi(P)$ quantifies the evaluation value at $\lambda$ randomly chosen parameter values from the set of candidate values $\Lambda$ of the parameter $v$ by $\phi(P,v)$ and returns the pair of the largest evaluation value and the parameter $v$ at that time as the value.

The main loop of the algorithm is as follows.

1. Determine one partition $P$ at random.

2. If the value of the evaluation function $\Phi(P)$ is less than $\rho$ or the number of iterations is less than $\gamma$, repeat $P = \psi(P, P, \sigma)$.

3. Return $P$ as the value.

The partition search function $\psi(P_0, P, \sigma)$ is a recursive function whose arguments are the initial partition $P_0$, the partition $P$, and the partition depth $\sigma$ described below. $\psi(P_0, P, \sigma)$ returns the partition whose evaluation function is greater than or equal to $\Phi(P_0)$.

1. If $\sigma$ is zero, return $P_0$ as the value.

2. Generate $P'$ from $P$ with the join partition modifier.

3. If $\Phi(P') > \Phi(P_0)$, return $P'$ as the value.

4. Generate $P''$ from $P$ with the separate partition modifier.

5. If $\Phi(P'') > \Phi(P_0)$, return $P''$ as the value.

6. let $P = \psi(P_0, P', \sigma\text{-}1)$.

7. If $\Phi(P) > \Phi(P_0)$, return $P$ as the value.

8. let $P = \psi(P_0, P'', \sigma\text{-}1)$.

9. If $\Phi(P) > \Phi(P_0)$, return $P$ as the value.

10. If none of the above conditions are satisfied, return $P_0$ as the value.

We apply this random search algorithm to the aforementioned search for enzymatic reaction networks with resettable bistability with the aim of demonstrating the effectiveness of the partition representation of enzymaric reaction networks. In other words, we design an evaluation function $\phi(P,v)$ for resettable bistability and use it in this algorithm.

## Results

### Evaluation function for resettable bistability

Bistability is the property of having two steady state values for a given parameter value $v$, as described above. Furthermore, resettable is the property that allows the transition between these two steady states by changing the value of the total concentration of the enzyme containing the input chemical species. Here, we create an evaluation function $\phi(P,v)$ that gives a high evaluation value for a typical resettable bistable aspect as shown in Fig 4A, i.e., an enzymatic reaction network that is monostable at the edge of the range taken by the total concentration of input chemical species and bistable in the central part.

$\phi(P,v)$ is a function of the partition $P$, with the reaction rate constants of each post-translational modification reaction and the total concentration of enzymes comprising the enzymatic reaction network corresponding to the partition as parameters $v$. The variations in the total concentration of enzymes containing the input chemical species were set to 21 discrete values of $\Omega = 2^{-5}, 2^{-4.5}, \ldots, 2^5$. These values, as well as the reaction rate constants to be set as $v$, are set to include approximately the values reported for the kinases that make up the MAPK cascade, which is known to be a typical cellular signaling system.

The following procedure corresponds to the aforementioned evaluation function $\phi(P,v)$. $P$ and $v$ are given as arguments.

1. For a given parameter value $v$, only the total concentration of the enzyme containing the input chemical species is varied sequentially and discretely for 21 values of $\Omega = 2^{-5}, 2^{-4.5}, \ldots,$ $2^5$, the total concentration of each enzyme is randomly assigned as the initial value so as to satisfy the conservation law of each enzyme. The steady state values were obtained by numerically solving the generated differential equations and making convergence judgments as appropriate. The details are described in the next section. Then 21 pairs of steady state values of the output chemical species are obtained. Here, the steady-state value of the output enzyme species is normalized to its total concentration, and is a value between 0 and 1.

2. This is repeated $\theta$ times to obtain $\theta$ pairs of normalized steady state values for each of the 21 different total concentrations of the input chemical species, where only data for which all 21 values are converged are used. Therefore, the number of them is at most $\theta$. Fixing one from 21 different total concentrations of the input chemical species means that we have at most $\theta$ steady state values. When all these values are the same, it means monostable, and when they are divided into two different values, it means bistable.

3. The degree of bistability can be quantitatively evaluated by the variance of several steady-state values obtained from different initial states. This value takes a maximum value of 1/4 in a perfect bistable state, i.e., when half of the values are 0 and the rest are 1, and a minimum value of 0 in a monostable state, i.e., when all the values are the same. It also takes an intermediate value in the mixed states. By multiplying the value of the variance by four, we can normalize the number representing the degree of bistability from 0 to 1.

4. Up to this point, we obtain 21 normalized variances $V = \{V_1, V_2, \ldots, V_{21}\}$, which indicate the degree of bistability for each of 21 values of the total concentration of the enzyme containing the input chemical species: $2^{-5}, 2^{-4.5}, \ldots, 2^5$.

5. Return the value of the function $eVL$ for this $V$, $eVL(V)$, as the evaluation value.

   The function $eVL$ is the function defined in Eq (10).

$$eVL(V) = \alpha \sum_{i=1}^{21} wVL(V_i, i)$$

$$wVL(v, l) = (v - |l - 11|/10)^2$$

(10)

Resettable bistability is the property of being monostable at the edge of the range taken by the total concentration of the input chemical species and bistable in the middle part, as explained in Fig 4. The $eVL$ evaluates the degree of this property. Its input $V$ is a vector of the above 21 quadruple variance values and $V_i$ denotes the each element. The value of the function $eVL$ was normalized by a maximum value so that it would be between 0 and 1. The $\alpha$ is the coefficient for this purpose, which is the reciprocal of the maximum value 0.604762 that the summation equation on the right side takes when $V = \{0, 0, 0, 0, 0, 0, 1, 1, 1, 1, 1, 1, 1, 1, 1, 0, 0, 0, 0, 0, 0\}$. The 3D shape of the function $wVL$ that makes up the function $eVL$ is shown in Fig 5. Its input $v$ is the quadruple variance value $V_i$ and $l$ is the position $i$ of $V_i$ in the vector $V$. The function $wVL$ is designed to be maximal for perfect monostability ($v = 0$) at the edge of the region, and also to be maximal for perfect bistability ($v = 1$) at the center of the region, and continuous intermediate evaluation values in the middle of the region.

## Hyperparameters of the search algorithm

The following is a summary of the hyperparameters included in the algorithm for searching for resettable bistable enzymatic reaction networks, which is an application of the partition

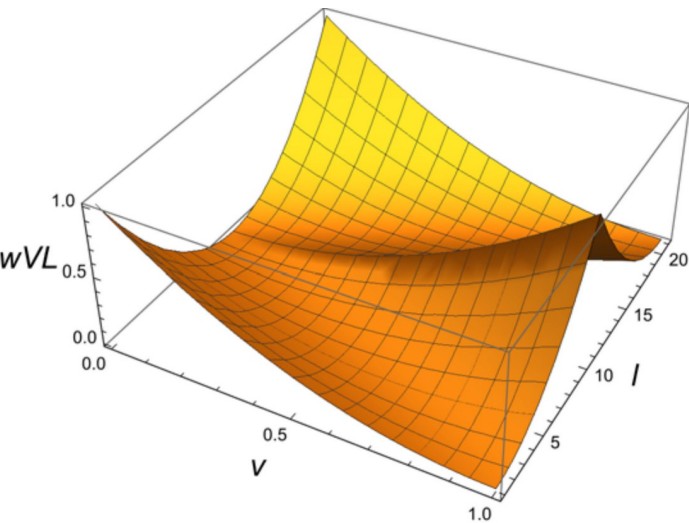

**Fig 5. Function *wVL* for evaluation of resettable bistability.** Input *v* is the quadruple variance value $V_i$ and *l* is the position *i* of $V_i$ in the vector *V*. The *wVL* is designed to take a maximum value for perfect monostability ($v = 0$) at the edge of the region, a maximum value for perfect bistability ($v = 1$) in the center, and continuous intermediate evaluation values in the middle of the region.

representation, and the values used in the actual search. First, the hyperparameters that are independent of the characteristics of the enzymatic reaction networks to be explored include the following.

- Number of post-translational modification reactions $N$: 4

- Input and output chemical species: input is $E_1$ and output is $P_2$

- Partitioning depth (depth of recursive search in binary tree) $\sigma$: 4

- Maximum number of loops to change the division $\gamma$: 150

- Value of the evaluation function to be judged as convergence $\rho$: 0.8

- A set of candidate values of a parameter to try randomly $\Lambda$: $\{2^{-5}, 2^{-4.5}, \ldots, 2^{5}\}$

- Number of pairs of parameter values to try randomly $\lambda$: 5

As mentioned earlier, the enzymatic reaction network shown in Fig 2B is known as a typical resettable bistable enzymatic reaction network [7], and the number of post-translational modification reactions that make up the network is 4, so the value of $N$ was set to 4. The input and output chemical species are set to be $E_1$ and $P_2$, respectively, which belongs to the different post-translational modification reactions, because the enzymatic reaction network explored would be severely limited if the input and output chemical species were those of the same post-translational modification reaction.

We did not know whether the values we chose for the depth of recursive search $\sigma$ and the maximum number of loops $\gamma$ were optimal or not. After trying several combinations, we chose the one that would allow us to explore a single enzyme reaction network in at least a few days. The set of values $\Lambda$ used for the total concentration of enzymes and the rate constants of the enzymatic reactions was set to include approximately the values in mmol/m$^3$-s system reported for the phosphatases comprising the MAPK cascade [21–25]. For $\lambda$, we tried several other values, but did not find much difference.

Hyperparameters specific to the search target, i.e., the search for bistable enzyme reaction networks, include the following. In general, these will alter for different search targets.

- A series of total concentrations of the input chemical species $\Omega$: $\{2^{-5}, 2^{-4.5}, \ldots, 2^{5}\}$

- Number of steady-state values with random initial values $\theta$: 5

We used the same variation of $\Lambda$ as described above for $\Omega$. We tried several other values, but did not find much difference for $\theta$.

To obtain the above steady state values, the generated differential equations were solved numerically many times while updating the initial values, and the convergence was judged. This part has the following hyperparameters, which are common when steady state values are needed in the search process.

- Calculation time for one step $\tau$: 10 s

- Maximum number of step repetitions $\kappa$: 360

- Convergence decision error $\varepsilon$: $10^{-3}$

Wolfram Mathematica's NDSolve function was used to solve the differential equations numerically. The protein concentration after $\tau$ seconds was calculated by NDSolve, and was judged to be converged when all of the relative error of the change, i.e., the ratio of the change to the total concentration of the protein, was less than $\varepsilon$. If there is no convergence, the protein concentration after another $\tau$ seconds is calculated and the same decision is made. If it does not converge after repeating this process $\kappa$ times, it returns the result that it did not converge. The product of $\tau$ and $\kappa$, 3600 seconds, or 1 hour, was taken as the critical time to reach steady state. This time was taken as a reference for the response time of the signaling system in the cell. The values of the calculation time $\tau$ for one step and the convergence decision error $\varepsilon$ were adopted after trial and error.

## Enzymatic reaction networks found by the search algorithm

The search was conducted on a machine with a 3.4GHz i7-3770 processor CPU, 16GB of memory, and a Windows 10 operating system. The search program was implemented in *Wolfram Mathematica* 12.0. The bistable enzymatic reaction networks found in the search at the values of the hyperparameters described above are shown in Fig 6A–6H. Each of them uses a different seed of random numbers for the initial setting. The maximum CPU time required to discover a single enzymatic reaction network was approximately 30 hours. For example, the enzymatic reaction network in Fig 6A consists of three chemical species: the input is the chemical species that identifies $E_1$, $E_2$, $P_1$, $P_4$, $S_1$, and $S_4$, and the output is the chemical species that identifies $P_2$ and $S_3$. The rest is a chemical species that identify $E_3$, $E_4$, $P_3$, and $S_2$. In particular, the input chemical species is in equilibrium with its dimer. Fig 6G is identical to the typical bistable enzymatic reaction network shown in Fig 2A. The reaction rate constants for each enzymatic reaction network, the total concentration of each chemical species, and the derived differential equation systems are shown in S1–S4 Tables provided as the supporting information, respectively.

Fig 7 shows the bistable aspect of each enzymatic reaction network, where a—h corresponds to a—h in Fig 6. In order to plot the bistable aspect, the Monte Carlo method was used to find the steady state values of the output chemical species for each value of the total concentration of the enzyme containing the input chemical species. That is, when numerically solving the system of differential equations derived from the partition, the initial values were randomly allocated to the chemical species included in each conservation law. We can see that all the

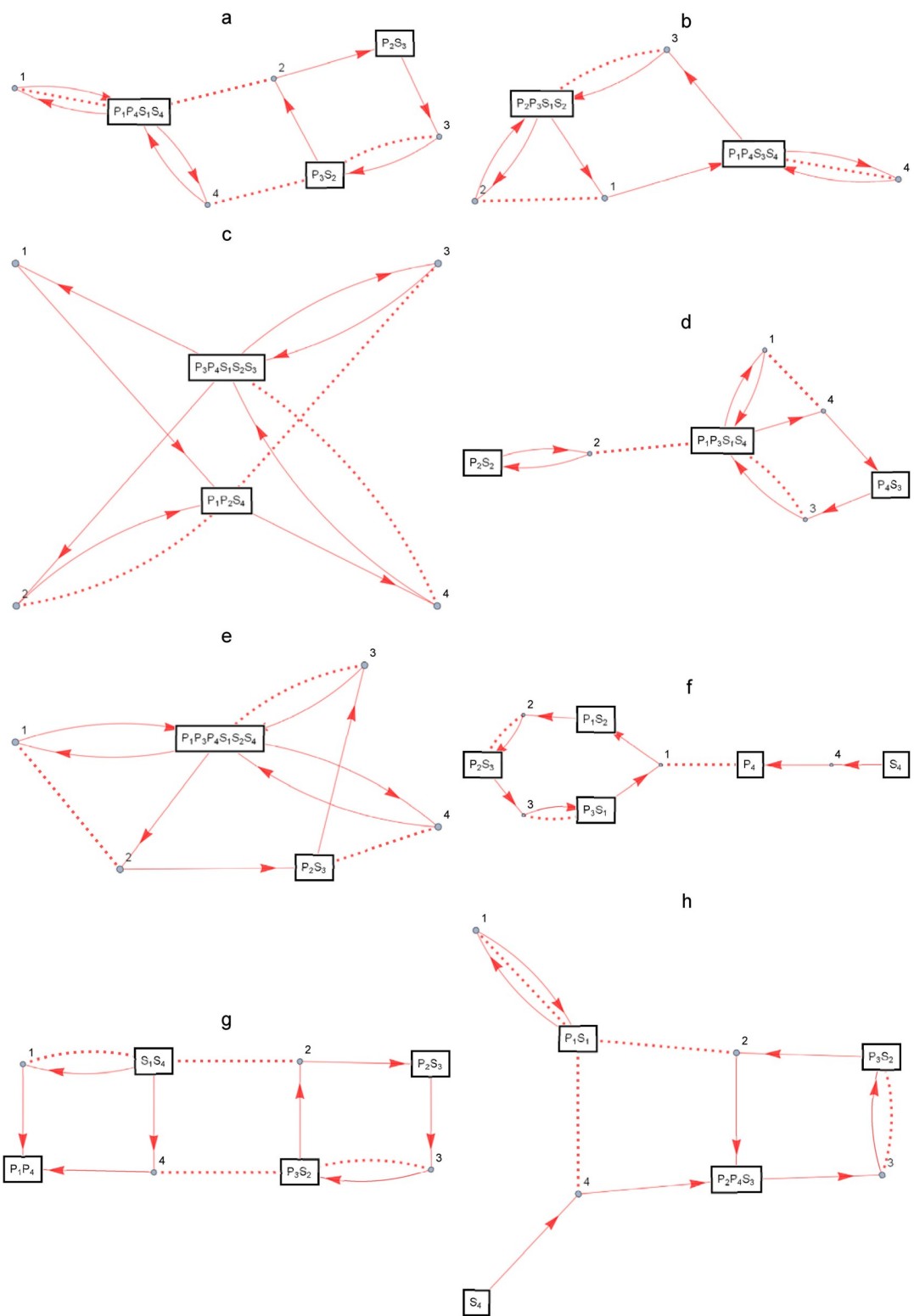

**Fig 6. Bistable enzymatic reaction networks found in the search.** The enzymatic reaction networks a—h use the different seeds of random numbers. The way to represent the figure is the same as in Fig 1.

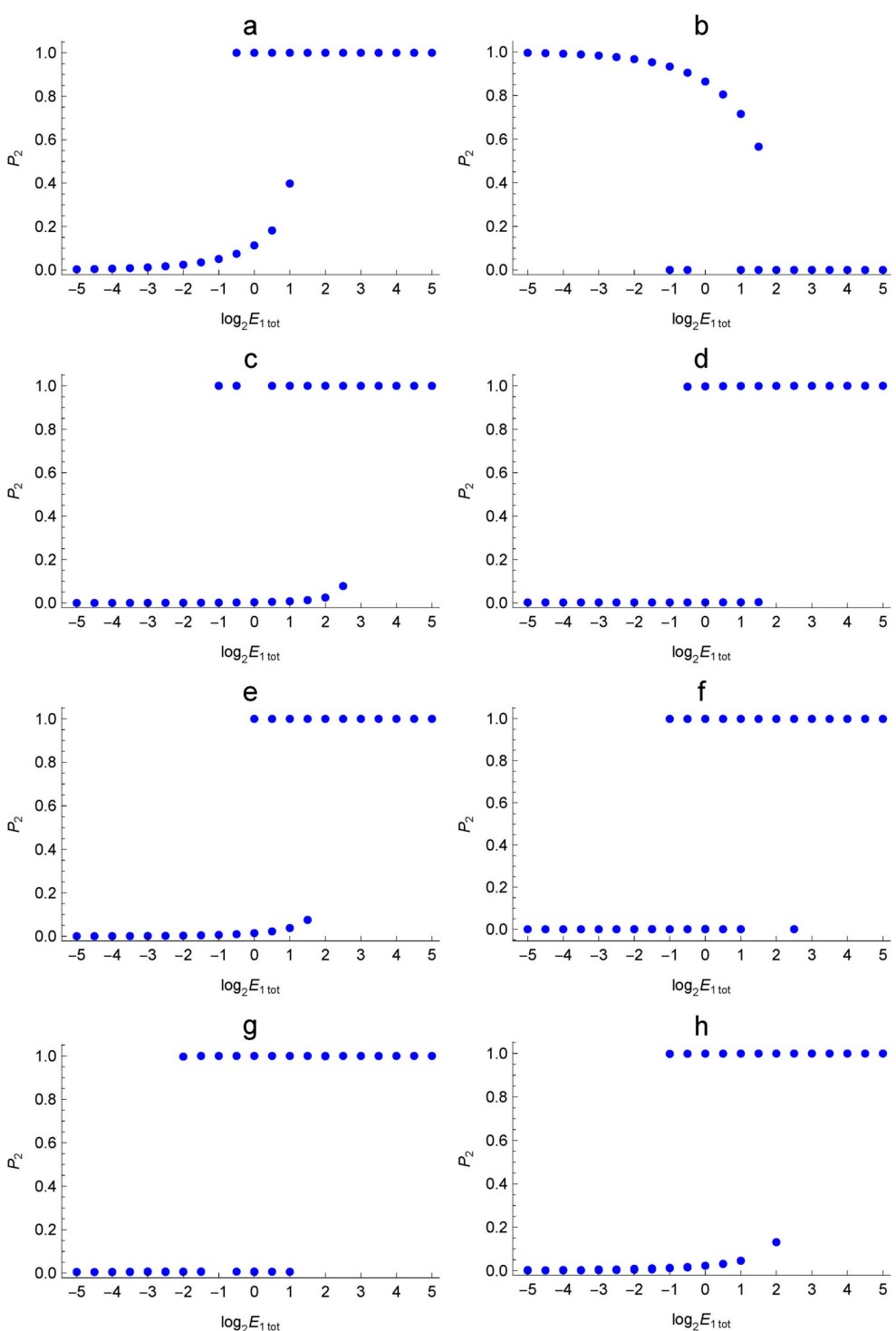

**Fig 7. Bistable aspects of enzymatic reaction networks found in the search.** The enzymatic reactoin networks a—h correspond to Fig 6. The horizontal axis is the total concentration of the enzyme including the input chemical species, $E_1$, in mmol/m³, and the vertical axis is the relative concentration of the output chemical species, $P_2$, to the total concentration.

obtained enzymatic reaction networks exhibit resettable bistability. Resettable is the property of being able to go back and forth between two bistable steady state values by changing the value of the input.

Fig 8 shows an aspect of the search process. Fig 8A shows the evolution of the evaluation value as the search progresses. The horizontal axis is the number of times the structure or parameter values of the enzymatic reaction network were changed. The vertical axis is the evaluation value, with the maximum being 1. The legends a—h corresponds to a—h in Fig 6. Since the evaluation value reaches 0.6 in all cases in the first few steps, and almost all graphs overlap up to that point, the range of the vertical axis is set to 0.55 or higher so that there is less overlap thereafter. The search is limited to 150 times, and the search is terminated when the evaluation value exceeds 0.8. It can be seen that the aspect of convergence varies depending on the seeds of random numbers used at initial settings. Fig 6B shows movements within the partition space,

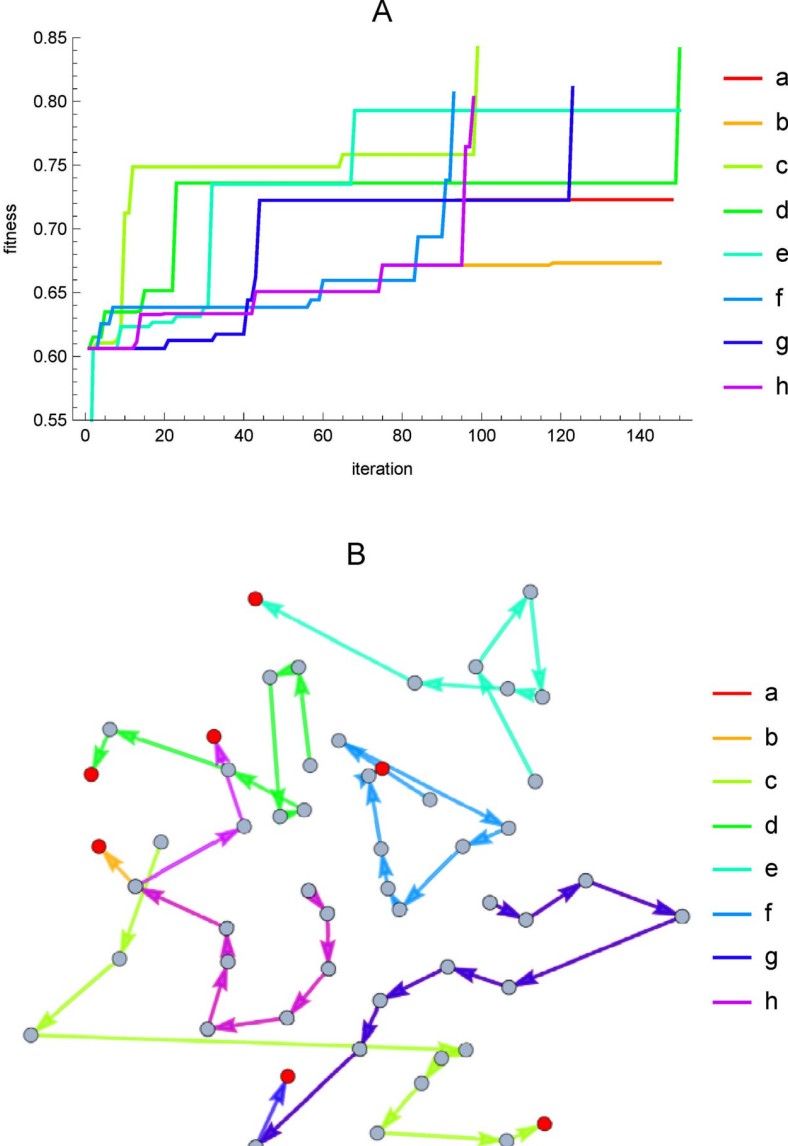

**Fig 8. Aspects of the search process.** The enzymatic reaction networks a—h correspond to Fig 6. A: Aspect of convergence. Horizontal axis is the number of iterations of the search. The vertical axis is the evaluation value. Convergence occurs when the evaluation value is greater than 0.8 or the number of iterations exceeds 150. B: Distribution of partitions in the search process based on the distance between partitions. The search proceeds in the direction of the arrow, and the red circles are the final points.

based on the distance between the partitions defined by Eq (3). Wolfram Mathematica's Graph function was used for drawing. The distance between all partitions was measured by the introduced distance function and the values were used in the EdgeWeight option. The GrpahLayout option specifies SpringEmbedding, which is arranged so that the total energy of the mutually coupled partitions is minimized by a spring of force equivalent to the distance between the partitions. The legends a—h corresponds to a—h in Fig 6. The red circles denote the final partitions in each search. We can see that the search is moving through the entire partition space.

## Discussion

We have shown that an enzymatic reaction network composed of post-translational modification reactions can be represented by a partition of the set whose elements are the enzymes, substrates, and products. By identifying elements of each subset in the partition as the same chemical species, an enzymatic reaction network can be constructed. In particular, by equating the substrate and product within the same post-translational modification reaction, we can also describe the association and dissociation reaction between enzyme and substrate (product). However, it does not represent the binding of multiple proteins as found in scaffold proteins and as seen in membrane receptor proteins. As an extension of the partition representation, enzyme-substrate complexes may be added as elements of the original set, but an algorithm for deriving the system of differential equations and conservation laws from the partition needs to be devised.

Furthermore, the partition representation of enzymatic reaction networks was applied to the search for bistable networks. The two partition modifiers introduced into the partition representation worked well and various bistable enzymatic reaction networks were obtained. One of the enzymatic reaction networks obtained in the search were the typical bistable enzymatic reaction network in which two cyclic reaction systems with positive auto-regulatory feedbacks mutually negatively regulate each other.

Regarding the performance of the search algorithm, it was confirmed that the search was completed within an acceptable processing time. It is especially noteworthy that the processing time required for the search was greatly reduced compared to an exhaustive search. In addition, the visualization of the search path using the distance between the partitions introduced in this study was able to show the aspect of the search.

It is not difficult to increase the granularity of the primary building blocks from post-translational modification reactions to cyclic reaction systems. Instead of the original set of elements being the substrate, enzyme, and product of the post-translational modification reaction, we can have four types of enzymes: the active enzyme, the inactive enzyme, the activating enzyme that catalyzes the reaction that turns the inactive enzyme into the active enzyme, and vice versa. We plan to try it in the future.

For the hyperparameter of the search algorithm, which is the search depth, and the parameter values for evaluating the structure of an enzymatic reaction network, which are the reaction rate constant and the number of times the total concentration of each chemical species is randomly tested, we set the search depth to 4 and the number of random tests to 5, but we did not systematically test other values. It is possible that adjusting these values can speed up the process. Further speed-up can be achieved by using GPUs or by using tQCCM [16], an improved form of the Michaelis-Menten approximation, as a formulation of the reaction mechanism instead of the law of mass action.

Although the bistability targeted in this study is a steady-state property, the search algorithm is divided into an evaluation function specific to the characteristics of the enzymatic reaction network to be explored and an independent algorithmic part, so that it may be possible to apply it to the exploration of dynamic properties such as biochemical adaptation, by

replacing the evaluation function. We plan to try it in the future. The number of post-translational modification reactions, which is the primary building block, was set to be 4, but even when the number was set to be 3, a bistable enzymatic reaction networks could be found. If the mechanism for automatic increase or decrease in the number of post-translational modification reactions can be included in the search algorithm, it will be possible to find a more optimal enzyme reaction network.

## Supporting information

**S1 Fig. An example of the transitions in search for enzymatic reaction networks.** Transitions of the enzymatic reaction networks in the process of searching for the discovered enzymatic reaction network Fig 6G. The top left is the initial enzymatic reaction network and transitions to the right, then moves to the second line and transitions from left to right, and the third line transitions from left to right as well. The bottom right is the final enzymatic reaction network Fig 6G.
(EPS)

**S1 Table. Reaction rate constant values for the enzymatic reaction networks found.** Each column is the value of the respective reaction rate constant. Each line corresponds to a—h in Fig 6.
(XLSX)

**S2 Table. The total concentration of enzymes in each enzymatic reaction network found.** Each line corresponds to a—h in Fig 6. The second column is a list of sets of chemical species corresponding to the conservation laws. The third column is their total concentration in the same order as the second column. However, for the set containing $E_1$, the input chemical species, we have moved it in the range of $2^{-5}$ to $2^5$, instead of the values in the table.
(XLSX)

**S3 Table. Differential equation system for each enzymatic reaction network found (a—d).** The first column corresponds to a—d in Fig 6. The second column is the name of the variable to be differentiated on the left side of the differential equation and the third column is the right side of the differential equation. The variable name is represented by x[chemical species name]. In particular, when some chemical species are considered identical, they are listed in brackets.
(XLSX)

**S4 Table. Differential equation system for each enzymatic reaction network found (e–h).** The first column corresponds to e–h in Fig 6. The second column is the name of the variable to be differentiated on the left side of the differential equation and the third column is the right side of the differential equation. The variable name is represented by x[chemical species name]. In particular, when some chemical species are considered identical, they are listed in brackets.
(XLSX)

**S1 Data.**
(ZIP)

## Author Contributions

**Conceptualization:** Takashi Naka.

**Data curation:** Takashi Naka.

**Formal analysis:** Takashi Naka.

**Funding acquisition:** Takashi Naka.

**Investigation:** Takashi Naka.

**Methodology:** Takashi Naka.

**Project administration:** Takashi Naka.

**Resources:** Takashi Naka.

**Software:** Takashi Naka.

**Supervision:** Takashi Naka.

**Validation:** Takashi Naka.

**Visualization:** Takashi Naka.

**Writing – original draft:** Takashi Naka.

**Writing – review & editing:** Takashi Naka.

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
