## [Decision Letter · Decision Letter 0]

15 May 2021

PONE-D-21-10250

The partition representation of enzymatic reaction networks and its application for searching bistable reaction systems

PLOS ONE

Dear Dr. Naka,

Thank you for submitting your manuscript to PLOS ONE. After careful consideration, we feel that it has merit but does not fully meet PLOS ONE’s publication criteria as it currently stands. Therefore, we invite you to submit a revised version of the manuscript that addresses the points raised during the review process.

Please note that both referee find the manuscript hard to follow and identify unsupported statements. We expect that the presentation of this work should be significantly improved before we can contact the referees again.

We look forward to receiving your revised manuscript.

Kind regards,

Ivan Kryven

Academic Editor

PLOS ONE

Journal Requirements:

Reviewers' comments:

Reviewer's Responses to Questions

**Comments to the Author**

1. Is the manuscript technically sound, and do the data support the conclusions?

Reviewer #1: No

Reviewer #2: Partly

2. Has the statistical analysis been performed appropriately and rigorously? 

Reviewer #1: No

Reviewer #2: N/A

3. Have the authors made all data underlying the findings in their manuscript fully available?

Reviewer #1: No

Reviewer #2: Yes

4. Is the manuscript presented in an intelligible fashion and written in standard English?

Reviewer #1: No

Reviewer #2: Yes

5. Review Comments to the Author

Reviewer #1: The paper is not intelligible the way it is currently written. It is difficult to say whether the findings merit publication if written better, because in the current form it is difficult to understand even what the main point is. The author says "proposed search algorithm worked well" in the abstract without specifying the basis for this statement. Similarly, there is a vague and general remark at the end of the abstract "seems to be applicable to the search for dynamic properties".

But mostly the main text of the paper is extremely unclear. Maybe the author can summarize the goal of the paper in a short paragraph, and explain one example clearly. Unfortunately, the way it is currently written, I am afraid the paper may be of little use to anyone who is not completely well-versed with the author's methods already.

Reviewer #2: The manuscript is about exploration of what kind of biochemical PTM networks can be bistable. The author came up with a clever approach for building networks from group-up using smaller pieces (partitions) as building blocks. I personally really appreciate this topic and would like to see this manuscript published. However, it requires a lot of work on explaining all the details of the approach.

Major isssues:

The concept of partitioning isn’t really clear. It isn’t thoroughly explained in the text, nor any references were provided. I searched this topic of partitioning enzymatic networks and found nothing. I assume this is the key novelty of the paper. So this piece really requires some additional work to make sure the concept is clear to the reader.

What is the point of Figure 1? Are these building blocks of partitioning? Is this a complete set of all building blocks?

Figure 2 also leaves a lot of questions. How does Fig 2a networks leads to {{P1,S1},{P1,S2,E1,E4 },{P4,S3},{P3,S4,E3,E2}} partitioning? Same pertains to even a more complicated Figure 2b. Are partitionings unique?

I’d suggest perhaps coming up with a figure that shows in details how the network is constructed from partitions.

Is the terms “partition” the right one? Usually this means that something was divided into smaller non-overlapping parts. I think the key concept of the paper is construction of the networks from smaller blocks. This slightly non-conventional use of terminology could be confusing for the reader.

Introduction of the distance in the partition space (equation 3) would also benefit from more explanations. I would suggest using a small example and show how the equation (3) can be applied. Evaluation of bistability described in Figure 3 isn’t clear either.

I wonder about the term “controllable bistability”. The author gives the definition and refers to the citation 20. Frankly, this is the first time I see the terms controllable bistability. Is there an uncontrollable one in the chemical reaction networks? Please provide an example of uncontrollable bistability so the use of “controllable” part is justified.

Results section, step 4 of the algorithm is the key for evaluating the bistability of the constructed network. What is the “degree” of controllable bistability? Then algorithm itself is kind of brief. Evaluation for bistability of a parameter-free network is a huge problem. I don’t fully understand what compromises have been taken in step 4, so the evaluation is quite short. It feels like it was a direct simulation of ODEs to reach the equilibrium and then concentration of output species plotted against the input. If this is the case, then it is a very risky approach. How to be sure that the hyperparameter space was searched exhaustively enough?

As I mentioned before, I’d love to see this published and used by the community. However, if the paper is published as is, it will be lost because it is hard to follow. If I may suggest an example of a paper that is written like an easy to follow tutorial is “Chemical Reaction Network Theory elucidates sources of multistability in interferon signaling” by Irene Otero-Muras et al, PLOS Comp Biol.

https://journals.plos.org/ploscompbiol/article?id=10.1371/journal.pcbi.1005454&rev=2

6. PLOS authors have the option to publish the peer review history of their article (what does this mean?). If published, this will include your full peer review and any attached files.

Reviewer #1: No

Reviewer #2: No

---

## [Author Response · Author response to Decision Letter 0]

24 Jun 2021

I revised the manuscript by taking account of the reviewer’s relevant comments. I tried to revise the manuscript with all my efforts and would like to give a point-by-point response to comments below. The reviewer’s comments are written in italics.

Answers to the comments by Reviewer #1:

The paper is not intelligible the way it is currently written. It is difficult to say whether the findings merit publication if written better, because in the current form it is difficult to understand even what the main point is. The author says "proposed search algorithm worked well" in the abstract without specifying the basis for this statement. Similarly, there is a vague and general remark at the end of the abstract "seems to be applicable to the search for dynamic properties".

The paper has been reviewed in general and additions have been made where necessary.

But mostly the main text of the paper is extremely unclear. Maybe the author can summarize the goal of the paper in a short paragraph, and explain one example clearly. Unfortunately, the way it is currently written, I am afraid the paper may be of little use to anyone who is not completely well-versed with the author's methods already.

The paper has been reviewed in general and additions have been made where necessary.

Answers to the comments by Reviewer #2:

The concept of partitioning isn’t really clear. It isn’t thoroughly explained in the text, nor any references were provided. I searched this topic of partitioning enzymatic networks and found nothing. I assume this is the key novelty of the paper. So this piece really requires some additional work to make sure the concept is clear to the reader.

The paper has been reviewed in general and additions have been made where necessary.

What is the point of Figure 1? Are these building blocks of partitioning? Is this a complete set of all building blocks?

The building block of partitions is only the post-translational modification reactions shown in Eq (1), and the N post-translational modification reactions are interconnected by species identification to form an enzymatic reaction network. The family of the set of chemical species to be identified is the partition representation. Fig 1 shows an example of the case where N=1 and N=2. In other words, Fig 1 is an example of an enzymatic reaction network composed of rather than building blocks.

To make the above clear, I added an example in Fig 1 for the case of N=1, and in the text, I added and modified the partition representations corresponding to each enzymatic reaction network. I also added and corrected the explanation of the network expression in the figure, which I think was missing.

Figure 2 also leaves a lot of questions. How does Fig 2a networks leads to {{P1,S1},{P1,S2,E1,E4 },{P4,S3},{P3,S4,E3,E2}} partitioning? Same pertains to even a more complicated Figure 2b. Are partitionings unique?

I’d suggest perhaps coming up with a figure that shows in details how the network is constructed from partitions.

As in Fig 1, the set M, which is the source of the partition, is explicitly shown for Fig 2. There is a one-to-one correspondence between the partition and the enzymatic reaction network that is constructed.

Is the terms “partition” the right one? Usually this means that something was divided into smaller non-overlapping parts. I think the key concept of the paper is construction of the networks from smaller blocks. This slightly non-conventional use of terminology could be confusing for the reader.

In this paper, partition is also used in the sense of a family of subsets, where the sets are disjoint to each other and their union set is the original set. In other words, it is a division into small non-overlapping parts. This is the representation of an enzymatic reaction network.

Introduction of the distance in the partition space (equation 3) would also benefit from more explanations. I would suggest using a small example and show how the equation (3) can be applied. 

Concrete examples were added to the explanation of the distance between divisions (Eq 3). In addition, diagrams of the enzymatic reaction networks corresponding to the examples were added as Fig 3.

Evaluation of bistability described in Figure 3 isn’t clear either.

The method of evaluating bistability using the function in Fig 3 (Fig 5 in the revised version) has been substantially added and revised.

I wonder about the term “controllable bistability”. The author gives the definition and refers to the citation 20. Frankly, this is the first time I see the terms controllable bistability. Is there an uncontrollable one in the chemical reaction networks? Please provide an example of uncontrollable bistability so the use of “controllable” part is justified.

In Reference 20, it is called “resettable bistability”. Thus, I will change to that term as well. Futhermore, I have added Fig 4 and a description to explain the relationship between bistability and resettability.

Results section, step 4 of the algorithm is the key for evaluating the bistability of the constructed network. What is the “degree” of controllable bistability? Then algorithm itself is kind of brief. Evaluation for bistability of a parameter-free network is a huge problem. I don’t fully understand what compromises have been taken in step 4, so the evaluation is quite short. It feels like it was a direct simulation of ODEs to reach the equilibrium and then concentration of output species plotted against the input. If this is the case, then it is a very risky approach. How to be sure that the hyperparameter space was searched exhaustively enough?

I have added an explanation to the method of evaluating controllable (resettable) bistability in step 4 of the algorithm.

Also, as you pointed out, as the stable equilibrium point (steady state) of the system, I used the value at which the time evolution of the system reached equilibrium in the direct simulation of the ODE. This is because the system is nonlinear and has a high order, so solving the algebraic equations corresponding to the steady state would eventually have to be done numerically, and determining whether the system is stable or unstable would require calculations such as finding the eigenvalues of the Jacobi matrix, which I thought would not be very efficient. I will add an organized explanation of the parameters and hyperparameters of the system in the text.

As I mentioned before, I’d love to see this published and used by the community. However, if the paper is published as is, it will be lost because it is hard to follow. If I may suggest an example of a paper that is written like an easy to follow tutorial is “Chemical Reaction Network Theory elucidates sources of multistability in interferon signaling” by Irene Otero-Muras et al, PLOS Comp Biol.

https://journals.plos.org/ploscompbiol/article?id=10.1371/journal.pcbi.1005454&rev=2

Thank you for introducing the appropriate literature. I will refer to it.

---

## [Decision Letter · Decision Letter 1]

15 Sep 2021

PONE-D-21-10250R1The partition representation of enzymatic reaction networks and its application for searching bistable reaction systemsPLOS ONE

Dear Dr. Naka,

Thank you for submitting your manuscript to PLOS ONE. After careful consideration, we feel that it has merit but does not fully meet PLOS ONE’s publication criteria as it currently stands. Therefore, we invite you to submit a revised version of the manuscript that addresses the points raised during the review process.

Please pay especial attention to the Referees' comments about the clarity of the algorithm used and the justification of its steps.

We look forward to receiving your revised manuscript.

Kind regards,

Ivan Kryven

Academic Editor

PLOS ONE

Journal Requirements:

Additional Editor Comments (if provided):

Reviewers' comments:

Reviewer's Responses to Questions

**Comments to the Author**

1. If the authors have adequately addressed your comments raised in a previous round of review and you feel that this manuscript is now acceptable for publication, you may indicate that here to bypass the “Comments to the Author” section, enter your conflict of interest statement in the “Confidential to Editor” section, and submit your "Accept" recommendation.

Reviewer #2: (No Response)

Reviewer #3: (No Response)

2. Is the manuscript technically sound, and do the data support the conclusions?

Reviewer #2: Partly

Reviewer #3: Yes

3. Has the statistical analysis been performed appropriately and rigorously? 

Reviewer #2: N/A

Reviewer #3: Yes

4. Have the authors made all data underlying the findings in their manuscript fully available?

Reviewer #2: No

Reviewer #3: Yes

5. Is the manuscript presented in an intelligible fashion and written in standard English?

Reviewer #2: No

Reviewer #3: No

6. Review Comments to the Author

Reviewer #2: While the author has addressed a lot of concerns, one of my major ones remains to be fully addressed. Specifically, I don’t understand how the evaluation function for resettable bistability (section starting on page 17, line 379) works. Either I am missing some details or this evaluation must be extremely computationally intensive.

First of all, my understanding is that the author decided to take a straightforward approach by evaluating the bistability by numerically solving the generated differential equations. Let’s say even in a fairly moderate biologically-relevant ODE system, in total there are about 10 total parameters (starting material amounts and kinetic constants). It seems like the author proposed to use 21 values per dimension. This translates into 16 trillion times solving the ODE system! Even if the number of dimensions is 5 (that limits the method to really primitive systems), it requires 4 million solutions. Let’s say one evaluation takes 1 second, this translates into about 1 month of computation time. So the straightforward numerical evaluation of bistability must be cursed by high dimensionality.

Again, for the sake of the benefit of the doubt, I may be missing something in the algorithm and/or the assumptions. I’d note that evaluation for resettable or not bistability is a hard problem. There are a variety of known approaches of how to evaluate bistability. Each of those approaches has its limitations. But no one in the field even considers evaluation of bistability for direct simulation of ODEs for the reasons outlined above.

The entire “Evaluation function for resettable bistability” needs to be improved.

1. Author needs to address the dimensionality of the problem.

2. There needs to be some discussion of kinetic constants along the species concentrations.

3. What is theta in step 2 (line 394)? How to estimate this theta?

4. Why does the author need to take quadruple of the variance? Why quadruple? What is the purpose of the entire step?

5. Since that point I’ve lost track.

Also, to make sure that the manuscript complies with "Have the authors made all data underlying the findings in their manuscript fully available?” question, I’d suggest making all the code and scripts available. I understand that the manuscript is mostly theoretical. However, if the description of the algorithms is not clear, the code may help.

Reviewer #3: This manuscript introduces a novel approach to explore dynamical properties of an enzymatic network. The novelty comes from the representation of an enzymatic network as a partition of the set instead of a regulatory matrix. This representation offers a possibility for an effective search of various dynamical properties of a network. The results demonstrate the exploration of reaction networks for the properties such as bistability and resettability.

The paper demonstrates a novel approach and important results for the exploration of bistable biological systems. However, the way this paper is written is not yet suitable for publication. “Introduction” and “Materials and Methods” sections were not comprehensible to me when reading the paper for the first time. The “Results” section clarified some confusion that appeared while reading the first part of the paper. Below are comments that might improve the readability of the paper and questions that appeared while reading it:

1. Resettability and bistability are rather central concepts in this paper. These terms appear a lot in all parts of the paper except in “Materials and Methods”. It seems like resettability and bistability drop out of that section and appear again in “Results”, making it harder for the reader to reconnect to the main story of the paper. I would recommend introducing these terms already in “Materials and Methods” next to enzymatic reaction networks which exhibit these properties. Figure 4a next to a visualization of an enzymatic reaction network which has this property would serve as a good illustrative example of what you are looking for. Then, the partition representation can be introduced for the purpose of finding networks with these properties.

2. In the subsection “Relationship between bistability and resettability” it is not clear which enzymatic network is discussed. “The relationship between bistability and resettability when the input chemical species is E1 and the output chemical species is P2 is shown in Fig 4” – it is not clear which network structure is discussed in this sentence and what are E1 and P2. Is it the network from Figures 1,2,3 or 6?

3. The algorithm descriptions in the section “Exploration of enzymatic reaction networks in the partition representation space” can be improved by either representing them as block diagrams or writing pseudo-code.

4. Section “Exploration of enzymatic networks in the partition representation space” can be framed with some intro paragraph explaining what to expect from the algorithm and some concluding paragraph summarizing the subsection. The section ends too abruptly with the steps of the algorithm. Moreover, it would create better connection with the rest of the paper if you already briefly mention how the search for resettability and bistability can be incorporated in this scheme.

5. Lines 51-53: “There, the control relationship between the cyclic reaction systems is represented by a control matrix whose elements are the cyclic reaction systems that make up the system.” – please rephrase. “Systems that make up a system” sounds confusing.

6. In the sentence in lines 75-78 you list various properties and only biochemical adaptation has a definition next to it. Please, make this consistent. You could introduce the definitions to all properties in a separate sentence, or even omit the definition and add a good reference for the interested reader.

7. Sentence in lines 79-82 has too many nested clauses, which makes it hard to grasp for the reader. Please, consider rephrasing it.

8. In the end of Introduction, term “resettability” appears rather unexpectedly. Moreover, ending the section with a definition worsens the readability of an article. Try to properly incorporate this term in the Introduction.

9. Line 93: please, rephrase the second half of the sentence and avoid “let’s think”.

10. Figures 1,2,3,6: rectangles for S and P would look better if they were white inside rather than transparent.

11. From line 124 till line 148 almost every other sentence starts with “In general”. Please, remove that or come up with another opening word.

12. Line 148: “dimer formation-separation reaction”

13. Line 163: “Fig 2 shows examples of more complex enzymatic reaction networks and its parftition representations” – typo in “partition”, and I believe it should be “reaction networks and their partition representations”.

14. Line 170: remove extra “the”.

15. Line 175-176: what do you mean by inactive and active MAPKKs? Maybe add some context for a reader who is not familiar with this system.

16. Line 185: “The advantage of the partition representation is that every partition is a representation…” – please, rephrase.

17. Line 197: “…the set M shown in Eq” -> “…the set M as shown in Eq”

18. Sentence in line 234: please expand or rephrase for better readability. The introduction of sk is a bit confusing.

19. Line 270: rk -> rk

20. Line 301: “…φ(P,ν) is a function of the partition P, with the parameter ν a pair of reaction…” – I believe you meant “… the parameter ν, which is a pair of reaction …”

21. Line 309: “and takes a maximum value among them as a value” -> “and takes a maximum among them as a value”. Line 309-310 – please, rephrase and reduce the number of words “value”.

22. Sentence in lines 317-320 is too long and might confuse the reader. If I understood it correctly, it can be split into 2 sentences: “The partition search function ψ(P0, P, σ ) described below is a recursive search function whose arguments are the initial partition P0, the current partition P, and the search depth σ. ψ(P0, P, σ ) returns the partition whose evaluation function value is greater than or equal to Ф(P0).”

23. Line 325: “is greater than of P0 ”

24. Line 330: the last step says “return P0 as a value” without any condition, which sounds confusing. Please add a condition, smth like “if all steps above are completed…” or “if none of the conditions above are satisfied…”

25. Line 336: “Resettablility is the property”. Resettable is an adjective.

26. Lines 371-372: “This is an example of a bistable that is not resettable” -> “This is an example of a bistable system (or a bistability?) that is not resettable”

27. The sentence in lines 508-513 is too long. Please, rephrase for better readability.

28. Figure 7 – why do some dots have different colors than others?

29. Figure 8 – as I understood, some arrows (like (a) and (g), and (b) and (h)) overlap significantly. Could you come up with a better representation? Maybe highlight it in the description of the figure or use dashed lines.

30. Lines 558-563 seem like an outlook. Consider first summarizing your findings and writing an outlook in the end of the Discussion. Also, line 563: I -> We, please, be consistent.

31. Line 569-570: “Good results” – please, avoid using this in a scientific paper.

7. PLOS authors have the option to publish the peer review history of their article (what does this mean?). If published, this will include your full peer review and any attached files.

Reviewer #2: No

Reviewer #3: No

---

## [Author Response · Author response to Decision Letter 1]

17 Oct 2021

Re: Letter to Editors

October 18, 2021

PLOS ONE

Editorial Office

Dear Editors:

Please find enclosed a revised manuscript entitled “The partition representation of enzymatic reaction networks and its application for searching bi-stable reaction systems” [PONE-D-21-10250R1] - [EMID d593f28512f8cd09], for publication in PLOS ONE.

I revised the manuscript by taking account of the reviewer’s relevant comments. I tried to revise the manuscript with all my efforts and would like to give a point-by-point response to comments below. The reviewer’s comments are written in italics.

Answers to the comments by Reviewer #2:

While the author has addressed a lot of concerns, one of my major ones remains to be fully addressed. Specifically, I don’t understand how the evaluation function for resettable bistability (section starting on page 17, line 379) works. Either I am missing some details or this evaluation must be extremely computationally intensive.

First of all, my understanding is that the author decided to take a straightforward approach by evaluating the bistability by numerically solving the generated differential equations. Let’s say even in a fairly moderate biologically-relevant ODE system, in total there are about 10 total parameters (starting material amounts and kinetic constants). It seems like the author proposed to use 21 values per dimension. This translates into 16 trillion times solving the ODE system! Even if the number of dimensions is 5 (that limits the method to really primitive systems), it requires 4 million solutions. Let’s say one evaluation takes 1 second, this translates into about 1 month of computation time. So the straightforward numerical evaluation of bistability must be cursed by high dimensionality.

Again, for the sake of the benefit of the doubt, I may be missing something in the algorithm and/or the assumptions. I’d note that evaluation for resettable or not bistability is a hard problem. There are a variety of known approaches of how to evaluate bistability. Each of those approaches has its limitations. But no one in the field even considers evaluation of bistability for direct simulation of ODEs for the reasons outlined above.

The entire “Evaluation function for resettable bistability” needs to be improved.

The method of constructing the evaluation function for resettable bistability has been modified to address each of the points pointed out below, and the overall description has also been modified to make it easier to understand.

1. Author needs to address the dimensionality of the problem.

The reviewer estimates that even if there are 5 parameters, if each value is 21 ways, we get 5^21 ≈ 4000000, which is equivalent to an exhaustive evaluation on the parameter values. I used to do this kind of exhaustive analysis, but as the reviewer pointed out, it is computationally explosive, so this is not the method used in this study. In this study, only five random combinations of parameter values are used for a single enzymatic reaction network, and the evaluation is done with those values. This value of 5 is the value of the parameter λ.

It should be pointed out, however, that the same enzymatic reaction network may be revisited, as we do not try to remove enzymatic reaction networks that have been evaluated once from the evaluation. Even in this case, the search depth σ of the two-branch search with two different partition modifiers is set to 4, so the maximum number of nodes in the binary tree is 62=2^1+2^2+2^3+2^4+2^5, and the maximum number of iterations of the main loop to change the partition is 150, so the computational cost is at most 9600=62*150 times. In addition, the actual number of parameters such as reaction rate constants and total concentration is not five, but four post-translational modification reactions, so the maximum number of reaction rate constants is 12=3*4 and the maximum number of total concentrations is 8=2*4, so the total number is a maximum of 20. This means that the cost of evaluation in the parameter space of one enzymatic reaction network is 5*9600 versus 21^20.

Why such a sparse sampling of parameter values can successfully reveal an enzymatic reaction network with the desired properties may require some explanation. Based on the exhaustive analysis I have done, one explanation is that the bistable nature of enzymatic reaction networks is dominated by the network structure and is quite robust in terms of changes in parameter values. This means that the bistable property is maintained over a quite wide range of parameter values. Alternatively, we can consider that a robust enzymatic reaction network is selectively found. However, it should be noted that it is not robust with respect to input chemical species, as we have placed the condition that it is resettable.

2. There needs to be some discussion of kinetic constants along the species concentrations.

The rate constants of the post-translational modification reactions and the total concentration of enzymes in the enzymatic reaction network are set to include approximately the values reported for the kinases that make up the MAPK cascade, which is known as a typical signaling system. This description was added in the revised text.

3. What is theta in step 2 (line 394)? How to estimate this theta?

The total concentration of each enzyme is randomly assigned as the initial value to satisfy the conservation law for each enzyme, and the process of finding 21 pairs of steady state values for the output chemical species is repeated theta times. As a result, we obtain theta steady states when the reaction rate constants of the constituent post-translational modification reactions and the total concentration of the enzyme are fixed. When all these values are the same, it means monostability, and when they are divided into two kinds of values, it means bistability. This explanation has been added to the relevant section. The value is set to 5 as appropriate, but as described in the discussion in the text, we tried other values and found no significant difference in the search results.

4. Why does the author need to take quadruple of the variance? Why quadruple? What is the purpose of the entire step?

From the theta steady state values obtained up to the previous step, the variance of those values can be used to quantitatively evaluate the degree of bistability. Since the steady state values are normalized in the range of 0 to 1, the variance takes a maximum value of 1/4 in a perfect bistable state, i.e., when half of the values are 0 and the rest are 1, and a minimum value of 0 in a monostable state, i.e., when all values are the same. So we multiply this value by 4 for the purpose of normalizing it. This explanation has been added to the corresponding step.

5. Since that point I’ve lost track.

As I answered in the beginning, I tried to revise the description to make it clearer for the other parts you pointed out.

Also, to make sure that the manuscript complies with "Have the authors made all data underlying the findings in their manuscript fully available?” question, I’d suggest making all the code and scripts available. I understand that the manuscript is mostly theoretical. However, if the description of the algorithms is not clear, the code may help.

For reference, the program code is submitted as supplementary material.

Answers to the comments by Reviewer #3:

This manuscript introduces a novel approach to explore dynamical properties of an enzymatic network. The novelty comes from the representation of an enzymatic network as a partition of the set instead of a regulatory matrix. This representation offers a possibility for an effective search of various dynamical properties of a network. The results demonstrate the exploration of reaction networks for the properties such as bistability and resettability.

The paper demonstrates a novel approach and important results for the exploration of bistable biological systems. However, the way this paper is written is not yet suitable for publication. “Introduction” and “Materials and Methods” sections were not comprehensible to me when reading the paper for the first time. The “Results” section clarified some confusion that appeared while reading the first part of the paper. Below are comments that might improve the readability of the paper and questions that appeared while reading it:

1. Resettability and bistability are rather central concepts in this paper. These terms appear a lot in all parts of the paper except in “Materials and Methods”. It seems like resettability and bistability drop out of that section and appear again in “Results”, making it harder for the reader to reconnect to the main story of the paper. I would recommend introducing these terms already in “Materials and Methods” next to enzymatic reaction networks which exhibit these properties. Figure 4a next to a visualization of an enzymatic reaction network which has this property would serve as a good illustrative example of what you are looking for. Then, the partition representation can be introduced for the purpose of finding networks with these properties.

Following your suggestion, I moved "Relationship between Bistability and Resettability" to " Materials and Methods" and changed it to "Resettable Bistability". I also referred to the enzymatic reaction network of Fig 2a as the example.

2. In the subsection “Relationship between bistability and resettability” it is not clear which enzymatic network is discussed. “The relationship between bistability and resettability when the input chemical species is E1 and the output chemical species is P2 is shown in Fig 4” ? it is not clear which network structure is discussed in this sentence and what are E1 and P2. Is it the network from Figures 1,2,3 or 6?

In "Resettable Bistability", which was moved and renamed in response to comment 1, I added an explanation referring to the enzymatic reaction network of Fig 2a as the example, and also added an explanation for E1 and P2 in that part.

3. The algorithm descriptions in the section “Exploration of enzymatic reaction networks in the partition representation space” can be improved by either representing them as block diagrams or writing pseudo-code.

I tried to break down each step of the algorithm and modify it to be at the level of pseudocode.

4. Section “Exploration of enzymatic networks in the partition representation space” can be framed with some intro paragraph explaining what to expect from the algorithm and some concluding paragraph summarizing the subsection. The section ends too abruptly with the steps of the algorithm. Moreover, it would create better connection with the rest of the paper if you already briefly mention how the search for resettability and bistability can be incorporated in this scheme.

Following your suggestion, I rearranged the description of the entire section and added the intro and final paragraphs.

5. Lines 51-53: “There, the control relationship between the cyclic reaction systems is represented by a control matrix whose elements are the cyclic reaction systems that make up the system.” ? please rephrase. “Systems that make up a system” sounds confusing.

I deleted the unnecessary part of the sentence you mentioned.

6. In the sentence in lines 75-78 you list various properties and only biochemical adaptation has a definition next to it. Please, make this consistent. You could introduce the definitions to all properties in a separate sentence, or even omit the definition and add a good reference for the interested reader.

I added explanations and references for all five properties mentioned.

7. Sentence in lines 79-82 has too many nested clauses, which makes it hard to grasp for the reader. Please, consider rephrasing it.

I broke the sentence you pointed out into two sentences to eliminate nesting, and also modified the next sentence slightly.

8. In the end of Introduction, term “resettability” appears rather unexpectedly. Moreover, ending the section with a definition worsens the readability of an article. Try to properly incorporate this term in the Introduction.

Resettability was introduced as an additional property of bistability, and an explanation was added.

9. Line 93: please, rephrase the second half of the sentence and avoid “let’s think”.

I paraphrased using "Considering".

10. Figures 1,2,3,6: rectangles for S and P would look better if they were white inside rather than transparent.

I followed your suggestion and whitened the inside of the rectangle.

11. From line 124 till line 148 almost every other sentence starts with “In general”. Please, remove that or come up with another opening word.

I’ve removed all "in general."

12. Line 148: “dimer formation-separation reaction”

I have corrected it as you suggested.

13. Line 163: “Fig 2 shows examples of more complex enzymatic reaction networks and its parftition representations” ? typo in “partition”, and I believe it should be “reaction networks and their partition representations”.

You are correct. I’ve corrected it.

14. Line 170: remove extra “the”.

I’ve removed the extra “the”.

15. Line 175-176: what do you mean by inactive and active MAPKKs? Maybe add some context for a reader who is not familiar with this system.

I followed your advice and added a brief description of the MAPK cascade.

16. Line 185: “The advantage of the partition representation is that every partition is a representation…” ? please, rephrase.

I paraphrased that sentence.

17. Line 197: “…the set M shown in Eq” -> “…the set M as shown in Eq”

I’ve fixed the sentence.

18. Sentence in line 234: please expand or rephrase for better readability. The introduction of sk is a bit confusing.

I rewrote the text, including the introduction of sk.

19. Line 270: rk -> rk

I’ve corrected the part you pointed out.

20. Line 301: “…φ(P,ν) is a function of the partition P, with the parameter ν a pair of reaction…” ? I believe you meant “… the parameter ν, which is a pair of reaction …”

You are correct. I’ve corrected the sentence.

21. Line 309: “and takes a maximum value among them as a value” -> “and takes a maximum among them as a value”. Line 309-310 ? please, rephrase and reduce the number of words “value”.

I revised the sentences and reduced the number of the word "value".

22. Sentence in lines 317-320 is too long and might confuse the reader. If I understood it correctly, it can be split into 2 sentences: “The partition search function ψ(P0, P, σ ) described below is a recursive search function whose arguments are the initial partition P0, the current partition P, and the search depth σ. ψ(P0, P, σ ) returns the partition whose evaluation function value is greater than or equal to Ф(P0).”

As you suggested, I split the sentence into two.

23. Line 325: “is greater than of P0 ”

I’ve fixed the sentence.

24. Line 330: the last step says “return P0 as a value” without any condition, which sounds confusing. Please add a condition, smth like “if all steps above are completed…” or “if none of the conditions above are satisfied…”

I have added the condition as you suggested.

25. Line 336: “Resettablility is the property”. Resettable is an adjective.

I’ve corrected the part you pointed out.

26. Lines 371-372: “This is an example of a bistable that is not resettable” -> “This is an example of a bistable system (or a bistability?) that is not resettable”

I’ve corrected the part you pointed out.

27. The sentence in lines 508-513 is too long. Please, rephrase for better readability.

I’ve corrected the part you pointed out to make it easier to read.

28. Figure 7 ? why do some dots have different colors than others?

The colors correspond to different initial values for obtaining the steady state values. However, there is no point in showing it in this figure, so we changed it to the same color.

29. Figure 8 ? as I understood, some arrows (like (a) and (g), and (b) and (h)) overlap significantly. Could you come up with a better representation? Maybe highlight it in the description of the figure or use dashed lines.

As you pointed out, some of the arrows overlap in Fig 8a. In particular, the evaluation value reaches 0.6 in all cases up to the first two steps, and almost everything else overlaps up to that point. So I changed the range of the vertical axis to 0.55 or higher, and modified it to reduce the overlap as much as possible after that. I added a note about this correction in the text and in the explanation.

30. Lines 558-563 seem like an outlook. Consider first summarizing your findings and writing an outlook in the end of the Discussion. Also, line 563: I -> We, please, be consistent.

Following your suggestion, I have revised the order to summarize the results first, and then the outlook at the end. The subject in line 563 has also been changed to "We".

31. Line 569-570: “Good results” ? please, avoid using this in a scientific paper.

I’ ve corrected the description to be more specific.

We hope the revised paper is of interest for the readers of PLOS ONE. I am looking forward to hearing from you again. 

Yours sincerely, 

Takashi NAKA

---

## [Decision Letter · Decision Letter 2]

13 Jan 2022

The partition representation of enzymatic reaction networks and its application for searching bistable reaction systems

PONE-D-21-10250R2

Dear Dr. Naka,

We’re pleased to inform you that your manuscript has been judged scientifically suitable for publication and will be formally accepted for publication once it meets all outstanding technical requirements.

Kind regards,

Ivan Kryven

Academic Editor

PLOS ONE

Additional Editor Comments (optional):

Reviewers' comments:

Reviewer's Responses to Questions

**Comments to the Author**

1. If the authors have adequately addressed your comments raised in a previous round of review and you feel that this manuscript is now acceptable for publication, you may indicate that here to bypass the “Comments to the Author” section, enter your conflict of interest statement in the “Confidential to Editor” section, and submit your "Accept" recommendation.

Reviewer #4: All comments have been addressed

2. Is the manuscript technically sound, and do the data support the conclusions?

Reviewer #4: Yes

3. Has the statistical analysis been performed appropriately and rigorously? 

Reviewer #4: Yes

4. Have the authors made all data underlying the findings in their manuscript fully available?

Reviewer #4: Yes

5. Is the manuscript presented in an intelligible fashion and written in standard English?

Reviewer #4: Yes

6. Review Comments to the Author

Reviewer #4: The work by Dr. Naka demonstrates a new computational approach to finding bistability in biochemical reaction networks. The work is technically sound and performed on an excellent professional level. The author adequately addressed comments from previous rounds of revision. I recommend this manuscript for publication at this stage.

7. PLOS authors have the option to publish the peer review history of their article (what does this mean?). If published, this will include your full peer review and any attached files.

Reviewer #4: No

---

## [Editor Report · Acceptance letter]

17 Jan 2022

PONE-D-21-10250R2 

The partition representation of enzymatic reaction networks and its application for searching bi-stable reaction systems 

Dear Dr. Naka:

I'm pleased to inform you that your manuscript has been deemed suitable for publication in PLOS ONE. Congratulations! Your manuscript is now with our production department. 

Kind regards, 

on behalf of

Dr. Ivan Kryven 

Academic Editor

PLOS ONE